# Safety and efficacy of very low carbohydrate diet in patients with diabetic kidney disease— A randomized controlled trial

**Nur Aisyah Zainordin**[1], **Nur' Aini Eddy Warman**[1], **Aimi Fadilah Mohamad**[1], **Fatin Aqilah Abu Yazid**[1], **Nazrul Hadi Ismail**[2], **Xin Wee Chen**[3], **Marymol Koshy**[4], **Thuhairah Hasrah Abdul Rahman**[5], **Nafeeza Mohd Ismail**[6], **Rohana Abdul Ghani**[1,7] *

1 Endocrine Unit, Dept of Internal Medicine, Faculty of Medicine, Universiti Teknologi MARA (UiTM), Shah Alam, Malaysia, 2 Centre for Dietetics Studies, Faculty of Health Sciences, Universiti Teknologi MARA (UiTM), Puncak Alam, Selangor, 3 Dept of Public Health Medicine, Faculty of Medicine, Universiti Teknologi MARA (UiTM), Shah Alam, Malaysia, 4 Dept of Radiology, Faculty of Medicine, Universiti Teknologi MARA (UiTM), Shah Alam, Malaysia, 5 Dept of Pathology, Faculty of Medicine, Universiti Teknologi MARA (UiTM), Shah Alam, Malaysia, 6 Dept of Pharmacology and Therapeutics, International Medical Universiti (IMU), Lumpur, Malaysia, 7 Institute of Pathology, Laboratory and Forensic Medicine (I-PPerForM), Universiti Teknologi MARA (UiTM), Shah Alam, Malaysia

* agrohana@gmail.com

## Abstract

### Introduction

There is limited data on the effects of low carbohydrate diets on renal outcomes particularly in patients with underlying diabetic kidney disease. Therefore, this study determined the safety and effects of very low carbohydrate (VLCBD) in addition to low protein diet (LPD) on renal outcomes, anthropometric, metabolic and inflammatory parameters in patients with T2DM and underlying mild to moderate kidney disease (DKD).

### Materials and methods

This was an investigator-initiated, single-center, randomized, controlled, clinical trial in patients with T2DM and DKD, comparing 12-weeks of low carbohydrate diet (<20g daily intake) versus standard low protein (0.8g/kg/day) and low salt diet. Patients in the VLCBD group underwent 2-weekly monitoring including their 3-day food diaries. In addition, Dual-energy x-ray absorptiometry (DEXA) was performed to estimate body fat percentages.

### Results

The study population (n = 30) had a median age of 57 years old and a BMI of 30.68kg/m2. Both groups showed similar total calorie intake, i.e. 739.33 (IQR288.48) vs 789.92 (IQR522.4) kcal, by the end of the study. The VLCBD group showed significantly lower daily carbohydrate intake 27 (IQR25) g vs 89.33 (IQR77.4) g, p<0.001, significantly higher protein intake per day 44.08 (IQR21.98) g vs 29.63 (IQR16.35) g, p<0.05 and no difference in in daily fat intake. Both groups showed no worsening of serum creatinine at study end, with consistent declines in HbA1c (1.3(1.1) vs 0.7(1.25) %) and fasting blood glucose (1.5(3.37)

**Data Availability Statement:** All relevant data are within the manuscript and its Supporting Information files.

**Funding:** RAG received partial research grants from the Malaysian Endocrine and Metabolic Society (MEMS)-LR1. NMI received partial research grant from the International Medical University (IMU)- IMU 418/2018. The funders had no role in study design, data collection and analysis, decision to publish, or preparation of the manuscript.

**Competing interests:** The authors have declared that no competing interests exist.

vs 1.3(5.7) mmol/L). The VLCBD group showed significant reductions in total daily insulin dose (39(22) vs 0 IU, p<0.001), increased LDL-C and HDL-C, decline in IL-6 levels; with contrasting results in the control group. This was associated with significant weight reduction (-4.0(3.9) vs 0.2(4.2) kg, p = <0.001) and improvements in body fat percentages. WC was significantly reduced in the VLCBD group, even after adjustments to age, HbA1c, weight and creatinine changes. Both dietary interventions were well received with no reported adverse events.

## Conclusion

This study demonstrated that dietary intervention of very low carbohydrate diet in patients with underlying diabetic kidney disease was safe and associated with significant improvements in glycemic control, anthropometric measurements including weight, abdominal adiposity and IL-6. Renal outcomes remained unchanged. These findings would strengthen the importance of this dietary intervention as part of the management of patients with diabetic kidney disease.

## Introduction

The current population of type 2 diabetes mellitus (T2DM) worldwide is over 400 million and increasing [1]. The prevalence of T2DM in Malaysia has approximately tripled over the last three decades, i.e. from 6.3% in 1986, to 17.5% in 2015 and most recently to 18.3% in 2019 [2]. T2DM is a progressive disease associated with debilitating microvascular and macrovascular complications. The national prevalence of chronic kidney disease (CKD) among the adult population in 2011 was reported as 9.1%. Among these, 4.2% patients were in CKD stage 1, 2.0% in CKD stage 2, 2.3% in CKD stage 3, whilst the rest were in stages 4 and 5 [3]. Furthermore, limited local data showed that diabetes is the leading cause of renal failure for patients commencing dialysis, with a steady rise from 53% of new patients in 2004 to 61% in 2013 [4]. Therefore, diabetic kidney disease (DKD) is a major T2DM complication that imposes debilitating health problems and significant financial burden on affected patients.

There has been an increasing amount of understanding of the complexity of the relationship between T2DM and obesity. As the prevalence of both conditions continue to demonstrate a parallel rise, the influence of obesity on T2DM is further marked. Thus, this has led to a greater emphasis on weight loss in the management of T2DM. Newer anti-diabetic medications, including SGLT-2 inhibitors and GLP1 agonists, are also gaining favor, not only for the greater efficacy in improving glycemic control but also due to the associated substantial weight loss [5]. Weight reduction programs are commonly complex and tedious with inconsistent, non-replicable and unpredictable outcomes, with emphasis on medical nutrition therapy and lifestyle changes [6, 7]. There had been many different dietary plans which share a common goal of reducing calorie intake in either the form of carbohydrate or fat, whilst increasing energy expenditure to achieve clinically meaningful weight loss. However, even weight management programs with structured modules to incorporate these elements produce small weight changes, often followed by weight regains, and commonly associated with high attrition rates [8–10].

Patients with DKD are known to have significantly higher cardiovascular risks in comparison to those with non-diabetic CKD, consequently demanding greater focus on risk reductions via pharmacological interventions as well as nutritional and dietary strategies. The focus on

dietary intervention in DKD has long been lowering protein consumption, with recommendations of less than 0.8 g/kg of body weight/day for non-dialysis dependent CKD, while higher dietary amount of more than 1.2 g/kg of body weight/day are recommended for patients on regular dialysis [11]. Despite recent interests in the impact of carbohydrate restriction in the management of T2DM [12], current knowledge on its benefits on diabetic nephropathy remain scarce. A recent systematic review concluded no connection between dietary carbohydrate content and serum creatinine and estimated glomerular filtration rate (eGFR) [13]. However, the ten studies included in the review enrolled patients with normal kidney function and any carbohydrate consumption restriction to less than 50g/day. The most recent study by Bruci et al demonstrated an effective and safe treatment with very-low-calorie ketogenic diet (VLCKD) in patients with obesity and mild kidney failure, including but not exclusive to those with T2DM [14].

Thus, to the best of our knowledge, this would be the first randomized controlled trial to study the safety and effects of very low carbohydrate (VLCBD) in addition to low protein diet (LPD) on renal outcomes in patients with T2DM and underlying mild to moderate kidney disease. The measured outcomes also included anthropometric, metabolic and inflammatory parameters.

## Research design and methods

This was an investigator-initiated, single-center, randomized, controlled, clinical trial in patients with T2DM and DKD, comparing 12-weeks of low carbohydrate diet to standard medical and nutritional therapy. Participants were recruited from the Endocrinology and Nephrology clinics of University Technology Mara Hospital Sungai Buloh, Malaysia. The duration of the study was from March 15th 2019 until March 5th 2021. The inclusion criteria were age 40–75 years old, diagnosis of T2DM of more than 5 years, stable chronic kidney disease defined by the CKD-EPI criteria of stages 2 or 3 [15], of more than 6 months, HbA1c between 7% - 10.5%, motivated to undergo the dietary intervention and able to provide informed consent. Patients were excluded if they had Type 1 diabetes mellitus, frequent hypoglycemia, had persistent elevations of serum transaminase, chronic heart failure (New York Heart Association functional class III-IV), active systemic inflammatory disease, chronic renal failure requiring dialysis, active hepatic and collagen diseases, malignancy, recent hospital admission within the past 3 months, pregnant, breastfeeding or planning to conceive within the next year. All patients who fulfilled the inclusion and exclusion criteria were invited to participate in the study and written informed consent was obtained. The principal investigators randomized the subjects via block randomization, based on HbA1c and gender to ensure equal numbers within both groups, and subsequent sealed envelopes, to dietary consultations of either very low carbohydrate plus low protein diet (VLCBD) or low protein diet only (LPD). All patients in both groups were given standard dietary and exercise advice, which included protein restriction to less than 0.8g/kg/day and low salt diet. Both investigator and subjects were not blinded to the interventions provided.

Patients within the VLCBD group were given a prescription diet of less than 20g of carbohydrate daily. This was supplemented by visual aids on carbohydrate counts of various local food. Patients were given the option to choose their most appropriate and palatable food types which would amount to the carbohydrate count given. Patients on oral anti-diabetic treatment, including insulin were advised on titrations of their medications to avoid hypoglycemia. The control arm was given standard low protein diet advice. All patients were required to fill in a 3-day food diary during their scheduled visits. Study visits were scheduled at 2-weekly intervals to assess dietary adherence based on the 3-day food diary, insulin dose adjustments referring

to home blood glucose monitoring and any hypoglycemic symptoms. However, due to the restriction movements imposed by the government during the study period, some of the visits were done via telecommunication, in which patients communicated with investigators and shared their food diaries for the dietary advice.

Clinical assessments, including blood samplings were done at baseline and at study end. Weight was monitored at each physical clinic visit. Blood and urine samples were collected after subjects had fasted overnight and were centrally analyzed at the Clinical Diagnostics Laboratory of the institution, an ISO 15189 accredited laboratory. Routine laboratory parameters such as creatinine, lipid profile [total cholesterol (TC), low-density lipoproteins (LDL-C), high-density lipoprotein cholesterol (HDL-C) and triglycerides (TG)], glucose and highly-sensitive C-reactive protein (hsCRP) were measured by an automated analyzer c501 (Roche Diagnostics, Germany), whilst interleukin-6 (IL-6) was run on the automated platform Cobas E411 (Roche Diagnostics, Germany). eGFR was calculated using the IDMS-traceable Modification of Diet Renal Disease (MDRD) equation.

Glycated hemoglobin (HbA1c) was analyzed on the semi-automated analyzer D-10 (Bio-Rad Laboratories Inc, USA), and values were expressed by using mmol/mol units according to the International Federation of Clinical Chemistry and Laboratory Medicine (IFCC) and were then converted to percentage values according to the National Glycohemoglobin Standardization Program (NGSP) by using the online HbA1c converter at http://www.ngsp.org/convert1.asp.

The study received ethical approval from the Research Ethics Committee of the institution (REC/448/18). The study is registered at ClinicalTrials.gov (NCT04931030).

## Dual Energy X-ray Absorptiometry scan (DEXA)

DEXA is an accurate, low radiation-emitting and non-invasive method that can measure bone mineral content (BMC), bone mineral density (BMD), fat mass and lean or fat-free mass (FFM). It works by measuring the attenuation of radiation beams passing through the body. Using low-energy x-ray intensities at two different energies, measurement of soft tissue masses (including fat and lean mass) and bone mineral is possible as fat and bone mineral have different attenuation properties. Not only does DEXA allow accurate and comprehensive measurements of total body fat percentage, but it also measures segmental body fat distribution in regions such as arms, legs, waist and hips. Aside from those mentioned above, it can also measure the visceral fat content in the abdomen. Whole-body composition analysis was performed with a Hologic Discovery DEXA system and analyzed using the manufacturer software. The measurements presented included total body fat (%), Est VAT Mass (g), Vol (cm3) and Area (cm2) [16].

## Study outcomes

The primary outcomes of the study were changes in creatinine, eGFR and urine microalbuminuria, between baseline values and at week 12 of the study. Secondary outcomes include changes of the following: (1) indexes of metabolic changes: weight, BMI, waist circumference (WC), hip circumference (HC), blood pressure and VAT (2) indexes of glycemic control: HbA1c and FBS levels (3) indexes of lipid metabolism: total cholesterol, HDL-cholesterol, LDL-cholesterol and triglyceride (TG) (4) changes in inflammatory markers: hsCRP and IL-6.

## Statistical analysis

Sample size calculation was based on a previous study, which showed an improvement in eGFR with LCBD after 6 months $69.0 \pm 14.5$ vs $69.4 \pm 15$, $P < 0.71$ and improvement of

UACR 141.7 ± 322.41 vs 96.8± 184.6 p< 0.69 [17]. Based on the assumption that LCBD will improve eGFR by 0.4 and UACR 200, the number of cases required to detect a significant difference in eGFR and UACR between the two groups under the conditions of two-sided $P$ value of 5% and power of 80%, was 16 patient per group, with a total sample size of 32. Considering 20% drop, we decided to enrol 20 patients in each arm for this study.

Data entry and analyses were done using SPSS version 27.0 (SPSS for Windows Version 27.0, SPSS Inc., Chicago, IL, USA). Only participants who completed the trials until week 12 were included in the analysis. The intervention (VLCBD) group was determined based on an as-treated basis. Participants' characteristics were presented in frequency, n (percentage, %) and median (interquartile range, IQR). For each participant, the change of the parameter measured was defined as the subtraction of parameter at week-12 and baseline, e.g., creatinine change for VLCBD group was obtained by subtracting the creatinine level post-intervention from the creatinine level pre-intervention. Their characteristic profiles and the parameter change. Their characteristic profiles and the parameter changes were compared using the Pearson Chi-Square or Fisher Exact test for categorical variables and Mann-Whitney U test (between-group analysis) or Wilcoxon signed-rank test (paired difference) for numerical variables. Analyses of the waist circumference effects over time was performed by a repeated measures Analysis of Variance (ANOVA), followed by a repeated measure of Analysis of Covariance (ANCOVA) using selected clinically important variables as the covariate(s). Cohen's (1988) cut-off points for the interpretation of correlation strength are used [18]. A p-value <0.05 is considered significant.

## Results

A total number of 38 patients consented to the study between March 2019 till December 2020. The group allocation was initially equal; however, due to the restrictions of clinic visits during the pandemic, disproportionate drop-out rate was recorded at Week-12, resulting in 16 subjects in the control group and 14 subjects in the intervention group to be included in the final analysis, which was completed in February 2021. However, there is no statistically significant drop between the groups. Refer **Fig 1**. The study cohort revealed medians in the age of 57 years old, duration of T2DM of 10.5 years and BMI of 30.68kg/m2. Median HbA1c was 8.8% with fasting glucose level of 8.45mmol/L, whilst median serum creatinine and eGFR were 104μmol/L and 59.83 mL/min/1.73m2, respectively. **Table 1** presented the individual characteristics of the participants. Both groups had comparable gender, age, duration of DM, and baseline parameters including systolic, diastolic blood pressure, and serum creatinine level, with the exception of total daily insulin use. Participants of the intervention arm had significantly lower total daily insulin doses compared to the control group (68 IQR 60 vs 124 IQR 94 IU, respectively, p 0.022).

Both groups demonstrated similar total daily calorie intake at baseline and week 12. Both groups also demonstrated similar total daily carbohydrate counts at baseline. At the 6-week and 12-week visits, the VLCBD group showed significantly lower carbohydrate counts in comparison to the control group, (week-6: 43.51 IQR 36.64 vs 105.67 IQR 54.41; week-12: 27.00 IQR 25 vs 89.33 IQR 77.4 g/day, p<0.001). However, the VLCBD group notably consumed higher daily intake of protein in comparison to LPD group at baseline, week 6 and week 12 intervals [baseline: 61.14 (IQR 38) vs 38.05 (IQR 22.75) g/day, p<0.01, week-6: 44.355 (IQR 22.52) vs 33.21 (IQR 20.16) g/day, p< 0.05, week-12 44.08 (IQR 21.98) vs 29.63 (IQR 16.35) g/day, p<0.05). Total fat intake in the VLCBD group was higher at baseline compared to LPD group [baseline: 45.93 (IQR 27.89) vs 33 (IQR 13) g/day, p = 0.028. However, both groups consumed similar total daily fat intake in the subsequent weeks. Refer Table 2.

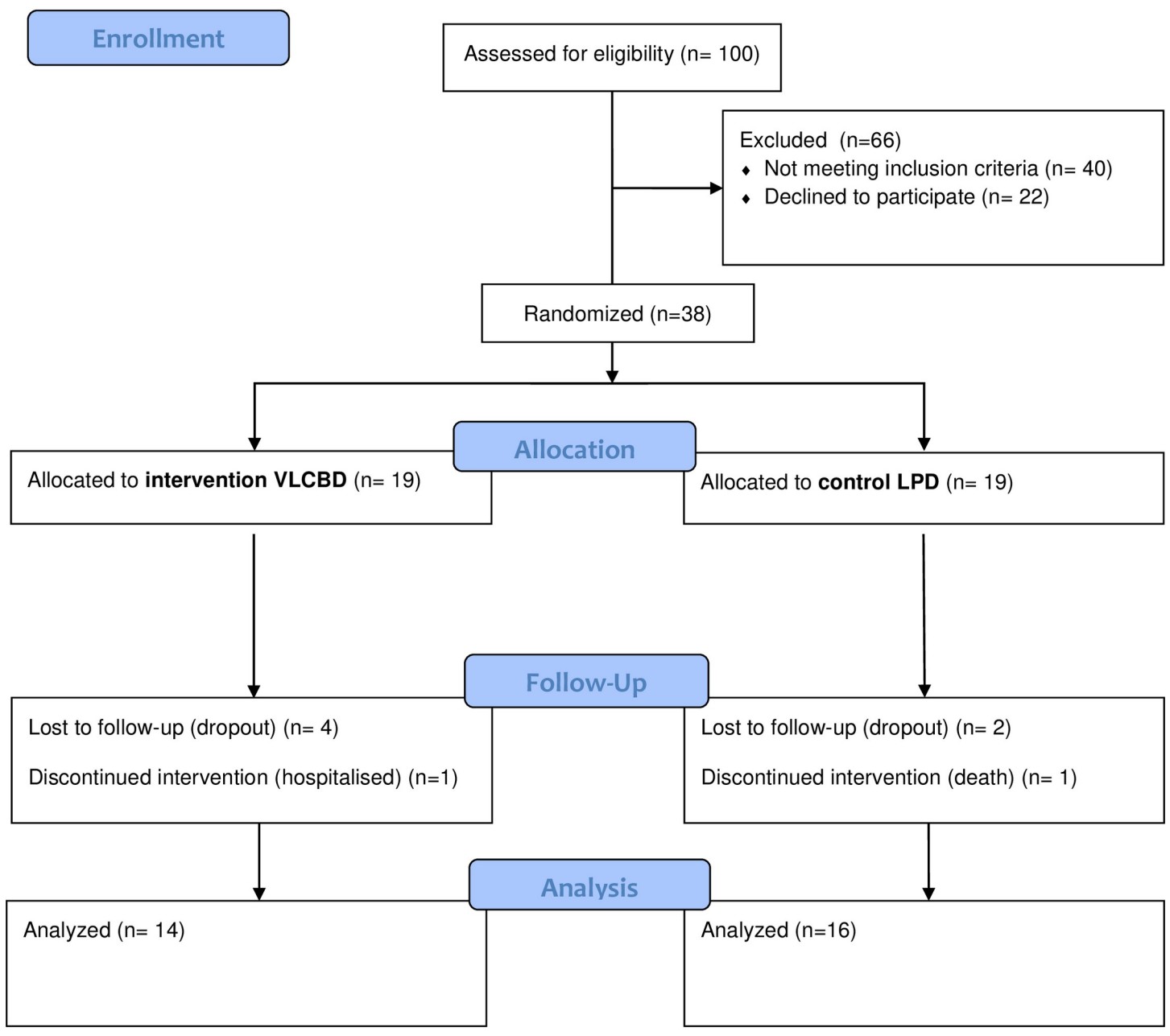

**Fig 1. Flow diagram for the overall study population.**

In regards to the renal outcomes, there were no statistically significant changes in the measured parameters, including no worsening of serum creatinine. However, we note a numerically large decline in the urine microalbuminuria within the VLCBD group (p = NS) whilst the control group demonstrated almost no change. Refer **Table 2**.

**Table 2** also revealed significant differences in medians of HbA1c, weight, and waist circumference between the two time periods. Twelve weeks of dietary advice resulted in both groups displaying significant reductions in HbA1c (p<0.05), which was numerically higher within the VLCBD compared to the LPD group (1.3 IQR1.1 vs 0.7 IQR1.25%, p = NS). This was accompanied by significant improvements in FPG within both groups, which was

**Table 1. Baseline demographic and clinical characteristics for whole populations (n = 30) and comparison between VLCBD and LPD groups.**

| Characteristic | Overall population | VLCBD group (n:14) | Control Groups (n:16) | p-value |
|---|---|---|---|---|
| Age, years | 57 (11) | 55 (13) | 57.5 (10) | 0.73 |
| Gender (male/female), n (%) | 18(60) /12 (40) | 7(50) /7 (50) | 11 (68.7)/5 (31.3) | 0.31 |
| Duration DM (years) | 10.5 (7) | 14.5 (10) | 10 (5) | 0.272 |
| Diabetic Medications, n(%) | | | | |
| Metformin | 27 (90) | 12 (40) | 15 (50) | 0.586 |
| Sulphonylureas | 3 (10) | 0 (0) | 3 (10) | 0.228 |
| DPP4-i | 17 (57) | 8 (26.7) | 9 (30) | 0.961 |
| SLGT2-i | 12 (40) | 7 (26.9) | 5 (19.2) | 0.431 |
| Insulin | 25 (83.3) | 12 (40) | 13 (43.3) | 0.743 |
| Medications | | | | |
| ACE-i/ARBs | 20 (66.7) | 9 (30) | 11 (36.7) | 0.796 |
| Calcium Channel | 13 (44.8) | 4 (13.8) | 9 (31) | 0.264 |
| Beta blocker | 7(23.3) | 3 (10) | 4 (13.3) | 0.818 |
| Diuretics | 6 (20) | 1 (3.3) | 5 (16.7) | 0.175 |
| Statin | 25(83.3) | 10 (33.3) | 15 (50) | 0.157 |
| Total Insulin Use, IU | 100(76) | 68 (60) | 126 (84) | **0.022** |
| Weight, kg | 83.85 (28.9) | 74.5 (23.2) | 89.65 (29.1) | 0.244 |
| BMI, kg/m2 | 30.68 (8.38) | 29.23 (7.26) | 32.21 (8.93) | 0.154 |
| Waist circumference, cm | 102.5(16.5) | 97 (16.8) | 112 (19) | 0.143 |
| Hip circumference, cm | 106 (14.8) | 107.5 (13.5) | 116 (29) | 0.321 |
| SBP, mmHg | 143 (21) | 145 (23) | 142 (21) | 0.868 |
| DBP, mmHg | 79.5 (17) | 79.5 (16) | 78.5 (19) | 0.617 |
| HbA1c, % | 8.8 (1.7) | 8.8 (1.7) | 8.55 (2.8) | 0.948 |
| Fasting plasma glucose, mmol/L | 8.45 (5.6) | 8.2 (2.9) | 9.0 (8.8) | 0.547 |
| Creatinine, mmol/L | 104 (43) | 97 (87) | 96 (37) | 0.967 |
| eGFR, mL/min/1.73 m2 | 59.83 (26.05) | 71.71 (47.24) | 65.0 (27.27) | 0.678 |
| UPCR, g/dL | 0.02 (0.06) | 0.02 (0.05) | 0.02 (0.08) | 0.718 |
| Urine microalbumin, mmol/L | 50.8(567) | 50.2 (655) | 50.8 (567) | 0.877 |
| LDL-C, mmol/L | 2.0 (1.0) | 2.05 (0.8) | 2.0 (1.2) | 0.868 |
| HDL-C, mmol/L | 1.1 (0.3) | 1.15 (0.4) | 1.05 (0.3) | 0.486 |
| Triglyceride, mmol/L | 1.95 (1.0) | 1.7 (0.9) | 2.20 (1.1) | 0.157 |
| Albumin, mmol/L | 44.5 (4.4) | 44.18±4.89 | 43.22±3.14 | 0.748 |
| ALT, mmol/L | 35.6 (25.9) | 26.05 (29.2) | 40.45 (23) | 0.280 |
| GGT, mmol/L | 52.5 (63) | 52 (62) | 52.5 (58) | 0.406 |
| ALP, mmol/L | 83.5 (40) | 83.5 (30) | 85.5 (48) | 0.708 |
| hsCRP, nmol/L | 2.81 (3.45) | 2.77 (5.34) | 2.81 (3.11) | 0.693 |
| IL-6, pg/mL | 3.88 (3.37) | 4.76 (3.17) | 3.37 (3.65) | 0.371 |
| Total body % fat | 35.5 (6.8) | 34.4 (6.5) | 37.2 (6) | 0.164 |
| Est VAT Mass (g) | 968 (264) | 953 (322) | 1002 (256) | 0.603 |
| Est VAT Vol(cm3) | 1046.5 (286) | 1030 (348) | 1083 (277) | 0.467 |
| Est VAT Area (cm2) | 200.5 (54.5) | 198.0 (66.5) | 208.0 (53.0) | 0.618 |

Data are presented as median (interquartile range) for continuous variables and percent for categorically distributed variables. P-values are results of Mann-Whitney test for continuous data for between the groups, and Fisher's exact test for categorical data. BMI; body mass index, DBP; diastolic blood pressure, HbA1c; hemoglobin A1c, SBP; systolic blood pressure, UPCR; urinary protein-to-creatinine ratio, HDL-C; high-density lipoprotein, LDL-C; low-density lipoprotein, ALT; Alanine aminotransferase, GGT; Gamma-glutamyl transferase, ALP; Alkaline phosphatase, hsCRP; high-sensitivity C-reactive protein, IL-6; Interleukin 6, Est VAT; Estimated visceral adipose tissue.

**Table 2. Comparison of changes in renal outcome, insulin requirement, biochemical and inflammatory markers between the VLCBD and LPD groups over 12 weeks of intervention.**

| Variable | | VLCBD group (n:14) | Control Groups (n:16) | *p-value* |
|---|---|---|---|---|
| **Total calories intake, kcal/day** | Baseline | 955.9 (377.42) | 965.42 (302.08) | 0.868 |
| | Week 6 | 727.75 (358.38) | 876.33 (309.88) | 0.967 |
| | Week 12 | 739.33 (288.48) | 789.92 (522.4) | 0.835 |
| | Change | -163.32 (528.48) | -77.33 (594.33) | 0.835 |
| | p-value within group | 0.084 | 0.079 | |
| **Total daily carbohydrate intake, g/day** | Baseline | 126.52 (48.24) | 124.02 (52.14) | 0.771 |
| | Week 6 | 43.51 (36.64) | 105.67 (54.41) | **<0.001** |
| | Week 12 | 27.00 (25) | 89.33 (77.4) | **<0.001** |
| | Change | -92.7(54.9) | -34.9(86.72) | **0.008** |
| | p-value within group | **<0.001** | 0.201 | |
| **Total protein intake, g/day** | Baseline | 61.14 (38) | 38.05 (22.75) | **0.01** |
| | Week 6 | 44.355 (22.52) | 33.21 (20.16) | **0.038** |
| | Week 12 | 44.08 (21.98) | 29.63 (16.35) | **0.012** |
| | Change | -14.14 (28.83) | -9.0 (27.67) | 0.739 |
| | p-value within group | 0.177 | 0.163 | |
| **Total fat intake, g/day** | Baseline | 45.93 (27.89) | 33 (13.3) | **0.028** |
| | Week 6 | 37.15 (25.28) | 32.2 (11.06) | 0.067 |
| | Week 12 | 35.225 (18.51) | 30.02 (18.67) | 0.146 |
| | Change | -9.28 (32.12) | -2.28 (22.35) | 0.647 |
| | p-value within group | 0.245 | 0.379 | |
| **Creatinine, mmol/L** | Baseline | 97 (87) | 96 (37) | 0.967 |
| | Week 12 | 105 (80) | 98 (48) | 0.917 |
| | Change | 4.0 (9.75) | 4.0 (13.25) | 0.967 |
| | p-value within group | 0.184 | 0.082 | |
| **eGFR, mL/min/1.73 m2** | Baseline | 71.71 (47.24) | 65.0 (27.27) | 0.678 |
| | Week 12 | 66.42 (38.82) | 65.65 (27.32) | 0.934 |
| | Change | -3.33 (8.97) | -2.63 (7.24) | 0.467 |
| | p-value within group | **0.046** | 0.173 | |
| **UPCR, g/dL** | Baseline | 0.02 (0.05) | 0.02 (0.08) | 0.718 |
| | Week 12 | 0.01 (0) | 0.02 (0.0) | 0.367 |
| | Change | 0.00 (0.01) | 0.00 (0.2) | 0.464 |
| | p-value within group | 0.832 | 0.178 | |
| **Urine microalbumin, mmol/L** | Baseline | 50.2 (655) | 50.8 (567) | 0.877 |
| | Week 12 | 25.5 (700) | 53 (418) | 0.407 |
| | Change | -3.2 (76.9) | 0.45 (67.1) | 0.477 |
| | p-value within group | 0.424 | 0.875 | |
| **Insulin, UI** | Baseline | 68 (60) | 124 (94) | **0.022** |
| | Week 12 | 30 (27) | 139 (96) | **0.001** |
| | Change | -39.0 (22) | 0.0 (0.0) | **<0.001** |
| | p-value within group | **0.002** | 0.285 | |
| **HbA1C, %** | Baseline | 8.8 (1.7) | 8.55 (2.8) | 0.948 |
| | Week 12 | 7.3 (1.7) | 8.15 (2.6) | 0.088 |
| | Change | -1.3 (1.1) | -0.7 (1.25) | 0.154 |
| | p-value within group | **.005** | **0.023** | |
| **FPG, mmol/L** | Baseline | 8.2 (2.9) | 9.0 (8.8) | 0.547 |
| | Week 12 | 6.3 (1) | 8.2 (5.5) | **0.162** |

(*Continued*)

**Table 2.** (Continued)

| Variable | | VLCBD group (n:14) | Control Groups (n:16) | *p-value* |
|---|---|---|---|---|
| | Change | -1.5 (3.37) | -1.3 (5.7) | 0.631 |
| | p-value within group | **.048** | **0.031** | |
| **LDL-C, mmol/L** | Baseline | 2.05 (0.8) | 2.0 (1.2) | 0.868 |
| | Week 12 | 2.40 (1.7) | 1.8 (0.9) | **0.024** |
| | Change | 0.55 (0.9) | -0.2 (0.5) | **0.001** |
| | p-value within group | **0.006** | 0.162 | |
| **HDL-C, mmol/L** | Baseline | 1.15 (0.4) | 1.05 (0.3) | 0.486 |
| | Week 12 | 1.2 (0.3) | 1.00 (0.2) | **0.015** |
| | Change | 0.05 (0.2) | -0.1 (0.1) | **0.016** |
| | p-value within group | 0.353 | **0.020** | |
| **Triglyceride, mmol/L** | Baseline | 1.7 (0.9) | 2.20 (1.1) | 0.157 |
| | Week 12 | 1.35 (0.9) | 1.75 (0.7) | 0.168 |
| | Change | -0.2 (0.75) | -0.25 (1.38) | 0.819 |
| | p-value within group | 0.172 | 0.178 | |
| **hsCRP, nmol/L** | Baseline | 2.77 (5.34) | 2.81 (3.11) | 0.693 |
| | Week 12 | 2.37 (3.76) | 3.29(5.92) | 0.803 |
| | Change | -0.31 (3.39) | 0.26 (1.85) | 0.101 |
| | p-value within group | 0.551 | 0.125 | |
| **IL-6, pg/mL** | Baseline | 4.76 (3.17) | 3.37 (3.65) | 0.371 |
| | Week 12 | 3.17 (2.51) | 3.74 (2.57) | 0.176 |
| | Change | -1.53 (3.35) | 0.46 (1.95) | **0.028** |
| | p-value within group | **.007** | 0.836 | |

Data are presented as median (interquartile range). P-values are results of Mann-Whitney test for continuous data for between the groups and Wilcoxon signed-rank test for two related samples/within the group; HbA1c; hemoglobin A1c, FPG; fasting plasma glucose, UPCR; urinary protein-to-creatinine ratio, HDL-C; high-density lipoprotein, LDL-C; low-density lipoprotein, hsCRP; high-sensitivity C-reactive protein, IL-6; Interleukin 6.

significantly lower in the VLCBD group at the end of the study. The VLCBD group showed a significantly greater reduction in total daily insulin dose compared to the control group (39 IQR 22 vs 0 IU, p<0.001). In regards to lipid profile, there was a significant small rise in LDL-C within the VLCBD group, whilst the LPD group demonstrated a smaller decline. The VLCBD group showed an increase in HDL-C of 0.05 IQR0.2 mmol/L compared to a decline in the LPD group of 0.1 IQR0.1 mmol/L, p = 0.02. TG levels remained unchanged.

The VLCBD group showed a significant reduction in IL-6 levels with a contrasting rise within the LPD group (-1.53 IQR3.35 vs 0.46 IQR 1.95, p = 0.028). Although the values were unable to reach statistical significance, there was a similar decline in hsCRP within the VLBCD group, which also seemed to challenge the rise in the LPD group (-0.314 IQR3.39 vs 0.26 IQR1.85, p = 0.101). Refer **Table 2.**

The VLCBD group demonstrated significant reductions in weight and BMI, which were not seen within the LPD group. Additionally, the median weight and BMI reductions were significantly greater within the VLCBD group compared to the control (-4.0 IQR3.9 vs 0.2 IQR4.2 kg, p = <0.001; -1.5 IQR1.18 vs 0.074 IQR1.54, p<0.001, respectively). This was accompanied by reductions in WC in both groups, which was numerically greater within the VLCBD group. The HC and the various DEXA body fat measurements showed consistent improvements within the VLCBD group, whilst the control group remained unchanged. The VLCBD group demonstrated a significant reduction in lean body mass, which were not seen in the LPD group (-2113.55 (IQR 1692.83) g vs 476.85 (IQR 2569.85) g, p <0.001). Refer **Table 3.**

**Table 3. Comparison of anthropometric and blood pressure changes between the VLCBD and LPD groups over 12 weeks of intervention.**

| Variable | | VLCBD group (n:14) | Control Groups (n:16) | *p-value* |
|---|---|---|---|---|
| **Weight, kg** | Baseline | 74.5 (23.2) | 89.65 (29.1) | 0.244 |
| | Week 12 | 72.5 (21.9) | 90.6 (29) | **0.042** |
| | Change | -4.0 (3.9) | 0.2 (4.2) | **<0.001** |
| | p-value within group | **0.002** | 0.955 | |
| **BMI, kg/m2** | Baseline | 29.23 (7.26) | 32.21 (8.93) | 0.154 |
| | Week 12 | 27.45 (6.34) | 32.15 (9.72) | **0.038** |
| | Change | -1.5 (1.18) | 0.074 (1.54) | **<0.001** |
| | p-value within group | **0.002** | 0.910 | |
| **Waist circumference, cm** | Baseline | 97 (16.8) | 112 (19) | 0.143 |
| | Week 12 | 95 (14.5) | 108 (18) | **0.006** |
| | Change | -4.0 (5.25) | -2.0 (3.6) | 0.167 |
| | p-value within group | **0.003** | **0.009** | |
| **Hip circumference, cm** | Baseline | 107.5 (13.5) | 116 (29) | 0.321 |
| | Week 12 | 103 (10.8) | 112 (23) | **0.008** |
| | Change | -5 (7.25) | 0.00 (6) | 0.056 |
| | p-value within group | **0.038** | .306 | |
| **SBP, mmHg** | Baseline | 145 (23) | 142 (21) | 0.868 |
| | Week 12 | 136 (30) | 137 (24) | 0.963 |
| | Change | -13 (31.75) | -11.0 (27.75) | 0.963 |
| | p-value within group | 0.315 | 0.158 | |
| **DBP, mmHg** | Baseline | 79.5 (16) | 78.5 (19) | 0.617 |
| | Week 12 | 80.00 (11.42) | 77.5 (11.02) | 0.662 |
| | Change | -1.5 (19) | -1.0 (11) | 0.963 |
| | p-value within group | 0.55 | 0.615 | |
| **Total body % fat** | Baseline | 34.4 (6.5) | 37.2 (6) | 0.164 |
| | Week 12 | 34.5 (6.3) | 35.5 (6.2) | 0.534 |
| | Change | -0.2 (2) | -1.4 (2.4) | 0.058 |
| | p-value within group | 0.420 | **0.005** | |
| **Est VAT Mass (g)** | Baseline | 953 (322) | 1002 (256) | 0.603 |
| | Week 12 | 812 (354) | 873 (379) | 0.300 |
| | Change | -201 (251.5) | -63 (219) | 0.16 |
| | p-value within group | **0.009** | .156 | |
| **Est VAT Vol(cm3)** | Baseline | 1030 (348) | 1083 (277) | 0.467 |
| | Week 12 | 878 (382) | 943 (409) | 0.137 |
| | Change | -218.0 (271.5) | -68 (236) | 0.160 |
| | p-value within group | **0.009** | 0.156 | |
| **Est VAT Area(cm2)** | Baseline | 198.0 (66.5) | 208.0 (53.0) | 0.618 |
| | Week 12 | 168.0 (73.5) | 181.0 (79.0) | .322 |
| | Change | -41.0 (51.95) | -13.0 (46) | 0.16 |
| | p-value within group | **0.008** | 0.156 | |
| **Lean Muscle Mass (g)** | Baseline | 46064.9 (14635.87) | 50224.7 (15734.2) | 0.383 |
| | Week 12 | 44310 (14324.38) | 50204.55 (17641.3) | 0.124 |
| | Change | -2113.55 (1692.83) | 476.85 (2569.85) | **<0.001** |
| | p-value within group | **0.002** | 0.148 | |

Data are presented as median (interquartile range) for continuous variables. P-values are results of the Mann-Whitney test for continuous data for between the groups and the Wilcoxon signed-rank test for two related samples/within the group. BMI; body mass index, DBP; diastolic blood pressure, HbA1c; hemoglobin A1c, SBP; systolic blood pressure, Est VAT; Estimated visceral adipose tissue.

**Table 4. Repeated measures ANCOVA (group*time-effect) comparing waist circumference between two different groups based on time (N = 30).**

|  |  | Mean difference (95% CI) | p-value* |
|---|---|---|---|
| WC (cm) | Unadjusted | 11.62 (3.097, 20.139) | **0.010** |
|  | Adjusted for age | 12.01 (2.987, 21.037) | **0.012** |
|  | Adjusted for HbA1c change | 11.63 (1.927, 21.335) | **0.021** |
|  | Adjusted for weight change | 14.62 (2.988, 26.242) | **0.016** |
|  | Adjusted for creatinine change | 11.70 (3.084, 20.319) | **0.010** |
|  | Adjusted for age and HbA1c change | 12.16 (1.919, 22.407) | **0.022** |
|  | Adjusted for age, HbA1c change and weight change | 14.99 (2.125, 27.860) | **0.025** |

CI, Confidence interval; WC, waist circumference

*Level of significance set at 0.05.

None of the patients reported any serious adverse events, including hospitalization for any cause, and there were no hypoglycemic events in both groups. Both dietary interventions were well received with no reported adverse events.

Repeated measures ANOVA analysis was performed to determine the within- and between-group effects. All models recorded an observed power of less than 80%. There was a significant mean WC difference (within-subjects effect) between the two time periods [mean diff: 3.43 (95% CI 2.067, 4.798), p<0.001]. The mean WC at baseline is higher than that at the 12th week. A significant difference in means of WC between the two time periods between two groups [mean diff: 11.62 (95% CI 3.097, 20.139), P = 0.010] was revealed, in which the mean WC in the control group is higher than that in the intervention group. Repeated measures ANCOVA between-group analysis with regard to time is shown in **Table 4,** which demonstrated that the effect of VLCBD on WC change (ranging from 11.62 cm to 14.99 cm) was consistently significant between the two time periods, with no evidence of an additive influence on the adjustment of the other risk factors, including age, HbA1c, weight and creatinine changes.

## Discussion

The present study involving a group of patients with obesity and T2DM showed that 12 weeks of very low carbohydrate intake in addition to standard protein restriction did not result in any worsening of renal outcome measurements. This was in contrast with previous reports that raised concerns on renal safety in low carbohydrate diets, primarily due to the compensatory rise in protein intake [19, 20]. Furthermore, the present study included subjects with underlying mild to moderate kidney disease, data from a population which is currently scarce, thus underscoring the potential benefit, albeit limited by lack of statistical significance, most likely due to the small sample size. However, we concur with the findings of Friedman *et al*, which demonstrated a 36% non-statistically significant reduction in albuminuria in a small group of patients with obesity and advanced diabetic nephropathy who received very-low-calorie ketogenic diet [21]. Although the intervention group was unable to achieve the targeted carbohydrate prescription of less than 20g a day, the median value of 27g per day was nonetheless substantially low. It was interesting, therefore, that a similar result was obtained in the present study with a more acceptable, less controversial and safe dietary prescription over a short duration of 12 weeks. The decline in eGFR in the VLCBD group is somewhat in agreement to the report by Ruggenenti *et al*, who concluded that calorie restrictions and subsequent weight loss could have conferred some renoprotection, particularly in those with glomerular

hyperfiltration [22]. This was also similar to a large observational study by Lin *et al*, who demonstrated a transient 10% decline in the eGFR within the first 3 months of follow up in a group of patients attending a weight management center, which subsequently plateaued over time [23]. We could only hypothesize that the lowering of eGFR by low calorie diet would have long term benefits of improving eGFR over time, as demonstrated by a previous report [24]. It is noteworthy, however, that this review and a more recent one, which recommended no relationship between low calorie diet and renal outcomes, were based on a population of patients with T2DM without renal impairment [13]. In addition, the reviews included studies with a low carbohydrate diet of less than 50g total intake a day and heterogenous in both duration as well as control groups. Another recent and similar study by Bruci *et al* possessed many similarities to our current study in regards to the low carbohydrate dietary intervention, 14 weeks in duration, and the study population of patients with obesity and mild kidney disease [14]. Despite the contrast in the comparator group (normal renal function), the establishment of ketosis, and the inclusion of patients with underlying chronic kidney disease, which was inclusive of, but not exclusive to, diabetic kidney disease, we are pleased to note that the study also demonstrated similar findings of safety and efficacy with the low carbohydrate diet of between 20-50g/day. Therefore, the present study has provided further evidence to highlight the possible benefit of very low carbohydrate dietary intervention, of almost 20g per day, without confirmed ketosis, in patients with underlying diabetic kidney impairment.

With regards to the baseline macronutrients intakes, the notably low baseline calorie consumption was probably be due to considerable under-reporting and under-estimation, both intentionally and unintentionally, as similarly reported by a previous study [25]. This could perhaps also explain the varying baseline protein and fat intakes between the 2 groups. The subsequent increase in the total daily protein intake in the VLCBD group was, however, an anticipated finding as patients attempt to compensate for the calorie restriction, as previously shown [22].

The VLCBD group exhibited an impressive mean weight loss of more than more than 4kg, which was an approximately 5.4% reduction from baseline, over a relatively short duration of 12 weeks. This was interestingly very similar to findings from previous short-term studies [22, 26]. This was evidently accompanied by reductions in waist circumference and further supported by significant reductions in estimated visceral fat mass, volume and areas as measured by the DEXA scan. Repeated measures ANOVA and ANCOVA between group analyses showed that WC reduction between two time periods was consistently significant, with or without adjustments of the other risk factors. These findings demonstrate robust evidences that severe carbohydrate restriction has a significant effect on reducing central obesity, which is subsequently linked to visceral adiposity. Thus, we strongly suggest that VLCBD has not only the advantage of significant weight loss but also the additional benefit of reducing visceral adiposity, which has been recognized as a significant independent predictor for metabolic and cardiovascular risks [27].

We are pleased to report a significant decline in HbA1c in both groups, which demonstrated that patients could be influenced to a certain degree by some form of dietary advice, as reported by Rolland, *et al* [28]. Notably, the VLCBD group demonstrated a greater improvement of more than 1%, compared to a median of 0.7% in the control group. This is a consistent finding that underscores the role of lowering dietary calorie content as a fundamental element in T2DM management [12]. The significant improvement in fasting glucose affirms the glycemic benefit. In addition, the VLCBD group demonstrated a decline in HbA1c to below 8%, which suggested a benefit on postprandial glucose levels as well. This is a significant finding as postprandial hyperglycemia has been previously shown to be a predominant contributor towards the development of visceral adiposity, leading to metabolic syndrome and

consequential cardiovascular risks [29]. Furthermore, this metabolic change could perhaps neutralize the seemingly negative impact of the increment in LDL-C within the group, which is consistent and replicated finding in current literature, frequently attributed to increased intake of saturated fat [30, 31]. However, it has been shown that the increase in LDL-C could be attributed to the formation of larger lipoprotein molecules which are less atherogenic [32]. The different changes in HDL-C were also worthy of mention. Albeit small, there was a significant reduction of HDL-C in the LPD group compared to a trend towards a rise in the VLCBD group. These changes are consistent with previous studies which reported an improvement in HDL-C with significant weight loss, particularly in low carbohydrate diet [31, 33]. Putting these findings into clinical perspectives, there is a clear need for physicians to address patients' lipid panels independently and providing adequate information to the patient of the potential consequences during a dietary advice particularly for low-carbohydrate diets.

IL-6 is a recognized inflammatory marker and has been utilized to represent chronic inflammation leading to cardiovascular disease [34]. The significant reduction in IL-6 within the VLCBD group was consistent with findings from Jonasson, *et al*, who concluded that low carbohydrate diet improved subclinical inflammatory state in T2DM as measured by various markers, including IL-6 [35]. Therefore, despite the elevation of LDL-C in the VLCBD group as discussed earlier, the decline in IL-6, accompanied by the minor rise in HDL-C, is perhaps more indicative of an overall reduction in the cardiovascular risks. hsCRP is one of the established surrogate markers for cardiovascular disease and has been incorporated as one of the factors for risk stratification [36]. We observed a median reduction in the VLCBD group, with an opposing rise in the LPD group, limited by the small sample size and thus lack of statistical significance. This somewhat concurred with the data from Ruth, *et al* who demonstrated a significant reduction in hsCRP in those who received high fat low carbohydrate diet versus those who received high carbohydrate diet [37]. There are a few plausible explanations for these positive findings. Lowering of HbA1c, as well as significant weight loss, have demonstrated improvements in inflammatory markers. Although it is almost impossible to examine this effect specifically on the dietary intervention, experimental research and population-based studies have demonstrated that high intake of refined or simple carbohydrates is associated with proinflammatory effects [38]. We consequently opine that the significant difference in the IL-6 changes between the 2 groups following the 12-week dietary interventions underscored the impact of significant carbohydrate restriction, particularly in this population of undisputedly high cardiovascular risk. However, as there is still scarcity of data in this area, further studies in a similar study population would be useful to affirm the findings.

This study had a relatively low attrition rate of 18%, considering the population was mainly among a group of middle-aged patients, particularly in the midst of the COVID-19 pandemic. We would like to emphasize this detail to reflect the feasibility of the dietary intervention, which included impositions by telecommunications via video calls due to the national movement restrictions. This, however, highlighted the fact that the labor-intensive dietary program could be eased by telecommunication visits with apparent successful clinical impact. The subjects had access to the diabetic educators and research assistants in the team to assist them in the event of any queries or untoward events. The present study also highlighted the practicality and efficacy of a weight management program among a group of men, which was in contrast to previous studies that suggested female participants tend to be more receptive and enthusiastic towards weight management programs compared to the opposing gender [39]. Future studies to identify significant confounding factors to influence motivation and adherence to the program would be useful.

We acknowledge that the study had a few limitations. The single-center data collection and the small number of participants made sub-group analyses difficult but notably could be

addressed in future studies. The COVID-19 pandemic halted further recruitment and imposed a lot of uncertainty for the expansion of the study. There was also a limitation in reviewing patient ketosis state in the intervention group as capillary and urinary ketones were not tested. In addition to that, only the estimated glomerular filtration was available as no direct kidney function measurements were obtained. Finally, although we were able to quantify the carbohydrate, protein and fat intakes, the study did not capture the saturated or unsaturated fat contents, which would be pursued in later studies. Furthermore, as food was not provided to the study participants, this could have resulted in under-reporting, which was somewhat addressed by using the 3-day food diary for each study visit. However, the available data presented here has provided some intriguing results, which we believe will encourage future studies in the areas of considerable carbohydrate restrictions in patients with T2DM and underlying kidney disease.

## Conclusion

In conclusion, to the best of our knowledge, this was the first study to determine the effect of very low carbohydrate dietary restriction, in addition to standard low protein diet in patients with underlying diabetic kidney disease. The intervention was safe with significant improvements in glycemic control, anthropometric measurements, including abdominal adiposity and IL-6. Renal outcomes were not affected. This would further support the growing data on the effectiveness of low carbohydrate diet as an important part of the management of T2DM, particularly in diabetic kidney disease.

## Supporting information

**S1 Checklist.**
(DOC)

**S1 File. The complete trial protocol has been provided as a supporting document.**
(PDF)

**S1 Dataset.**
(SAV)

## Acknowledgments

We would like to thank all the medical and non-medical personnel involved in the data collection and patient management.

## Author Contributions

**Conceptualization:** Nafeeza Mohd Ismail, Rohana Abdul Ghani.

**Data curation:** Nur' Aini Eddy Warman, Aimi Fadilah Mohamad, Fatin Aqilah Abu Yazid, Nazrul Hadi Ismail, Marymol Koshy.

**Formal analysis:** Xin Wee Chen, Rohana Abdul Ghani.

**Funding acquisition:** Nafeeza Mohd Ismail, Rohana Abdul Ghani.

**Investigation:** Thuhairah Hasrah Abdul Rahman.

**Methodology:** Nazrul Hadi Ismail, Rohana Abdul Ghani.

**Supervision:** Rohana Abdul Ghani.

**Validation:** Nazrul Hadi Ismail, Marymol Koshy, Thuhairah Hasrah Abdul Rahman.

**Writing – original draft:** Nur Aisyah Zainordin, Rohana Abdul Ghani.

**Writing – review & editing:** Marymol Koshy, Rohana Abdul Ghani.

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
