## [Decision Letter · Decision Letter 0]

29 Jul 2021

PONE-D-21-17598

Safety and Efficacy of Very Low Carbohydrate Diet in Patients with Diabetic Kidney Disease

PLOS ONE

Dear Dr. Abdul Ghani,

Thank you for submitting your manuscript to PLOS ONE. After careful consideration, we feel that it has merit but does not fully meet PLOS ONE’s publication criteria as it currently stands. Therefore, we invite you to submit a revised version of the manuscript that addresses the points raised during the review process.

The Reviewers asked for more details about the intervention and raised some significant concerns about the statistical methodology. Please address all the comments raised in your response.

We look forward to receiving your revised manuscript.

Kind regards,

Dorit Samocha-Bonet

Academic Editor

PLOS ONE

2. Please ensure that the registry number for this clinical trial is reported in the main text of the manuscript;

3. In your Methods section, please ensure that sample size calculations are reported, and that main and secondary outcomes are clearly listed, together with the duration of the trial. Also, please describe the location of the study.

Additional Editor Comments (if provided):

Reviewers' comments:

Reviewer's Responses to Questions

**Comments to the Author**

1. Is the manuscript technically sound, and do the data support the conclusions?

Reviewer #1: Yes

Reviewer #2: Partly

2. Has the statistical analysis been performed appropriately and rigorously? 

Reviewer #1: Yes

Reviewer #2: No

3. Have the authors made all data underlying the findings in their manuscript fully available?

Reviewer #1: Yes

Reviewer #2: Yes

4. Is the manuscript presented in an intelligible fashion and written in standard English?

Reviewer #1: Yes

Reviewer #2: Yes

5. Review Comments to the Author

Reviewer #1: Zainordin et al provide interesting data on the impact of a VLCKD on diabetic kidney disease, reporting that, compared to a low protein diet, no change in kidney function was observed, but a significant loss in visceral fat together with weight loss and better glucose control were achieved in the study group.

General comments:

English language needs some revision. Please use people first language throughout the manuscript

Keywords: suggest to use the acronym vlckd as vlcbd is not established

Abstract

Please provide more detail on calorie and macronutrient content of the two diets in the abstract.

Introduction:

please specify to which nation the national prevalence of t2d refers to.

Methods:

why such a small age bracket?

Patients had a mean BMI compatible with obesity, which is a well established risk factor for CKD and diabetes. Moreover, it is well-known that weight loss improves CKD as the authors report in the discussion. Was the low protein diet in keep with the guidelines for obese diabetic patients with stage 2-3 CKD?

Results and discussion

Were capillary or urinary ketones tested? If not please mention this in the limitations.

Please also mention that only the estimated filtration is available, and no direct kidney function measure was obtained.

Was the drop out rate significantly different across groups? If so, please elaborate the finding in the discussion

"The main investigators randomized the subjects via block randomization, based on HbA1c and gender to ensure equal numbers within both groups, and subsequent sealed envelopes, to

dietary consultations of either very low carbohydrate plus low protein diet (VLCBD) or

low protein diet only (LPD). All patients were given the standard dietary and exercise

advice which included protein restriction to less than 0.8g/kg/day and low salt diet."

It is somewhat unclear throughout the manuscript whether the low carb group also received low protein and low salt advise. Please clarify whether this was the case or not. A low protein diet which is also very low in carbs is necessarily either extremely hypocaloric or very high in fats. Moreover, if fats are not regulated by a diet, they often are saturated. Please provide data on the calorie, protein and fat (quantity as well as quality) consumed at baseline, week 6 and 12, so to understand which is the case. Please elaborate the findings accordingly.

Please provide muscle mass change

Another study investigating the impact of a very low carbohydrate hypocaloric ketogenic diet on patients with mild kidney failure did not report significant alterations in renal function long term (10.3390/nu12020333). Please complete the introduction/discussion commenting these results.

Reviewer #2: My 1st major concern is that the results do not seem correct in Table 2 and 3. For example, the changes in WC seems to be switched between VLCBD and control in Table 3. The within-group differences of medians between baseline and week are 4 and 2 for VLCBD and control respectively. But its shows a reduction of 4 in VLCBD and 2 in control. The similar issue happens to other outcomes in Table 2 and 3 also. Perhaps it is better to present full dataset in supplement table or using dot plots for those primary outcomes.

The sample size is too small to compare so many outcomes (more than 20 and 4 tests for each outcome). P value needs to be adjusted for so many tests.

The analyses and results in Table 4 are confusing.

Details:

1. Remove z values from all tables.

2. Table 1. For categorical variables (e.g. medications), add % also in each group. The numbers for medications do not seem to aligned correctly. If you did not use Wilcoxon signed-rank test for Table 1, remove it from the footnote.

3. Table 2. Some results are questionable:

a. Total carbohydrate intake: what are the median within group change?

b. Creatinine: control group seems to have an increase of 2 instead of 4.

c. Urine microalbumin: the decrease of VLCBD seems to be 25 (50.2-25.5), not -3.2

d. Insulin, UI: the increase in control seems to be 15 not 0.

e. LDL: z score of VLCBD should be positive instead of negative

4. Table 3:

a. Waist circumference : it seems that the reduction in VLCBD is 2 and the reduction in control is 4.

b. Hip: the reduction in control seems to be 4 instead of 0.

c. DBP: week 12 used “+-” instead of parentheses. Change misspelled as “hange”.

d. Est VAT Mass: the reduction in control seems to be 130 instead of 63.

e. Est VAT Vol: the reduction in control seems to be 140.

f. Est VAT Area: the reduction in control seems to be more than 27.

5. Table 4 is confusing.

a. Since interactions between group and time were included, it is better to show whether the interactions are significant. If the interactions are significant, then need to show simple effects (i.e. group differences at baseline and at week 12 separately). If the interactions are not significant, then you can show main effects.

b. Be careful when WC is included in the model for weight. There is obvious collinearity between group and WC change. The significant p value (0.031) is questionable if WC change is included.

c. Showing so many different adjustments is confusing to the readers. A model selection is preferred and show the best clinical meaningful/ data fit model as the final model.

d. It may be better to adjust for baseline WC and weight in the models.

6. PLOS authors have the option to publish the peer review history of their article (what does this mean?). If published, this will include your full peer review and any attached files.

Reviewer #1: No

Reviewer #2: No

---

## [Author Response · Author response to Decision Letter 0]

16 Aug 2021

We would like to thank both Reviewers for their constructive comments and suggestions. We hereby provide the responses to the questions and comments.

Reviewer #1: Zainordin et al provide interesting data on the impact of a VLCKD on diabetic kidney disease, reporting that, compared to a low protein diet, no change in kidney function was observed, but a significant loss in visceral fat together with weight loss and better glucose control were achieved in the study group.

General comments:

English language needs some revision. Please use people first language throughout the manuscript

- Thank you for the comment. The manuscript has been thoroughly revised to correct any typing or grammatical errors and to use people first language throughout the manuscript.

Keywords: suggest to use the acronym vlckd as vlcbd is not established

- Thank you for the suggestion. However, we would like to maintain the use of VLCBD as oppose to VLCKD, if the reviewer would allow us, as the present study did not confirm ketosis as did Bruci et al. 

Abstract

Please provide more detail on calorie and macronutrient content of the two diets in the abstract.

- Thank you for the comment. Details on calorie and macronutrient contents have been included.

Introduction:

please specify to which nation the national prevalence of t2d refers to.

- Thank you for the question. “The prevalence of T2DM in Malaysia has approximately….” has been inserted.

Methods:

Why such a small age bracket?

- Thank you for the question. We had decided to keep the age range small to maintain homogeneity in the study population. Future studies would be looking at slightly more diverse populations in age and levels of eGFR.

Patients had a mean BMI compatible with obesity, which is a well-established risk factor for CKD and diabetes. Moreover, it is well-known that weight loss improves CKD as the authors report in the discussion. Was the low protein diet in keep with the guidelines for obese diabetic patients with stage 2-3 CKD?

- Thank you for the question. The low protein diet is consistent with the current recommendations from local guidelines for non-dialysis CKD patients as stated in the Introduction section. To the best of our knowledge, none of the current guideline discriminate patients with normal BMI or those with obesity. However, we have added statements to reflect the actual protein intakes in the populations both in the Result and Discussion sections.

Results and discussion

Were capillary or urinary ketones tested? If not please mention this in the limitations.

Please also mention that only the estimated filtration is available, and no direct kidney function measure was obtained.

- Thank you for the suggestions. No capillary or urinary ketones were not measure. These have been included in the limitation section.

Was the drop out rate significantly different across groups? If so, please elaborate the finding in the discussion.

- Thank you for the question. We have added “However, there is no statistically significant drop between the groups.” in the Result section Para 1. 

"The main investigators randomized the subjects via block randomization, based on HbA1c and gender to ensure equal numbers within both groups, and subsequent sealed envelopes, to dietary consultations of either very low carbohydrate plus low protein diet (VLCBD) or low protein diet only (LPD). All patients were given the standard dietary and exercise advice which included protein restriction to less than 0.8g/kg/day and low salt diet."

It is somewhat unclear throughout the manuscript whether the low carb group also received low protein and low salt advise. Please clarify whether this was the case or not. 

- Thank you for the question and we apologise for the ambiguity. The statement has been modified to – “All patients in both groups were given standard dietary and exercise advice which included protein restriction to less than 0.8g/kg/day and low salt diet.”

A low protein diet which is also very low in carbs is necessarily either extremely hypocaloric or very high in fats. Moreover, if fats are not regulated by a diet, they often are saturated. Please provide data on the calorie, protein and fat (quantity as well as quality) consumed at baseline, week 6 and 12, so to understand which is the case. Please elaborate the findings accordingly.

- Thank you for the comments. The results of calorie, protein and fat consumption have been included in the Result section Para 2, Pg 12. This has been elaborated on in the Discussion section, Para 2, Pg 20. However, we are unable to provide the data on quality as this was not captured in the data collection. We have included this in the limitation.

Please provide muscle mass change.

- Thank you for the suggestion. This has been added in Result Para 5, Pg 15. “The VLCBD group demonstrated significant reductions in lean body mass which were not seen in LPD group ( -2113.55 (IQR 1692.83) g vs 476.85 (IQR 2569.85) g, p <0.001).”

Another study investigating the impact of a very low carbohydrate hypocaloric ketogenic diet on patients with mild kidney failure did not report significant alterations in renal function long term (10.3390/nu12020333). Please complete the introduction/discussion commenting these results.

- Thank you for the suggestion. Bruci et al reported very interesting results. This has been added accordingly in the Introduction section final paragraph and cited as Reference no 14. It has also been included in Discussion section, para 1. 

References numbering has been adjusted accordingly.

Reviewer #2: My 1st major concern is that the results do not seem correct in Table 2 and 3. For example, the changes in WC seems to be switched between VLCBD and control in Table 3. The within-group differences of medians between baseline and week are 4 and 2 for VLCBD and control respectively. But its shows a reduction of 4 in VLCBD and 2 in control. The similar issue happens to other outcomes in Table 2 and 3 also. Perhaps it is better to present full dataset in supplement table or using dot plots for those primary outcomes.

The sample size is too small to compare so many outcomes (more than 20 and 4 tests for each outcome). P value needs to be adjusted for so many tests.

The analyses and results in Table 4 are confusing.

Details:

1. Remove z values from all tables.

2. Table 1. For categorical variables (e.g. medications), add % also in each group. The numbers for medications do not seem to aligned correctly. If you did not use Wilcoxon signed-rank test for Table 1, remove it from the footnote.

- Thank you for the suggestions. Amendments have been made accordingly. 

3. Table 2. Some results are questionable:

a. Total carbohydrate intake: what are the median within group change?

- Thank you for the question. The results have been added. 

b. Creatinine: control group seems to have an increase of 2 instead of 4.

c. Urine microalbumin: the decrease of VLCBD seems to be 25 (50.2-25.5), not -3.2

d. Insulin, UI: the increase in control seems to be 15 not 0.

- Thank you for the question. However, the data expressed in the tables here were generated by the SPSS statistical analyses software program, with values expressed as median (IQR). Therefore, the data were not simple deductions. The full dataset has been uploaded in a Supplementary File as required. 

e. LDL: z score of VLCBD should be positive instead of negative

- This has been removed as suggested earlier. 

4. Table 3:

a. Waist circumference : it seems that the reduction in VLCBD is 2 and the reduction in control is 4.

b. Hip: the reduction in control seems to be 4 instead of 0.

c. DBP: week 12 used “+-” instead of parentheses. Change misspelled as “hange”.

d. Est VAT Mass: the reduction in control seems to be 130 instead of 63.

e. Est VAT Vol: the reduction in control seems to be 140.

f. Est VAT Area: the reduction in control seems to be more than 27.

- Thank you for the question. However, the data expressed in the tables here were generated by the SPSS statistical analyses software program, with values expressed as median (IQR). Therefore, the data were not simple deductions. 

- Lean Muscle Mass changes have been added in the Table 

- The full dataset has been uploaded in a Supplementary File as required. We apologize for the typing errors and these had been corrected accordingly.

5. Table 4 is confusing.

a. Since interactions between group and time were included, it is better to show whether the interactions are significant. If the interactions are significant, then need to show simple effects (i.e. group differences at baseline and at week 12 separately). If the interactions are not significant, then you can show main effects.

b. Be careful when WC is included in the model for weight. There is obvious collinearity between group and WC change. The significant p value (0.031) is questionable if WC change is included.

c. Showing so many different adjustments is confusing to the readers. A model selection is preferred and show the best clinical meaningful/ data fit model as the final model.

d. It may be better to adjust for baseline WC and weight in the models.

- Thank you for your comments. The amendments have been done by our co-author accordingly. Some explanations are included in the text as well as the modifications of the table as suggested. 

Thank you again for your suggestions. We look forward to your positive reviews.

---

## [Decision Letter · Decision Letter 1]

25 Aug 2021

PONE-D-21-17598R1

Safety and Efficacy of Very Low Carbohydrate Diet in Patients with Diabetic Kidney Disease

PLOS ONE

Dear Dr. Abdul Ghani,

Thank you for submitting your manuscript to PLOS ONE. After careful consideration, we feel that it has merit but does not fully meet PLOS ONE’s publication criteria as it currently stands. Therefore, we invite you to submit a revised version of the manuscript that addresses the points raised during the review process.

The reviewers are mostly satisfied with the revised manuscript. Please comment on what seems to be common underreporting of energy intake in the discussion, as per Reviewer 1. If food was not provided to the study participants, please add as a limitation as well as the dietary reporting using 3-days diet diaries.  

We look forward to receiving your revised manuscript.

Kind regards,

Dorit Samocha-Bonet

Academic Editor

PLOS ONE

Journal Requirements:

Reviewers' comments:

Reviewer's Responses to Questions

**Comments to the Author**

1. If the authors have adequately addressed your comments raised in a previous round of review and you feel that this manuscript is now acceptable for publication, you may indicate that here to bypass the “Comments to the Author” section, enter your conflict of interest statement in the “Confidential to Editor” section, and submit your "Accept" recommendation.

Reviewer #1: All comments have been addressed

Reviewer #2: All comments have been addressed

2. Is the manuscript technically sound, and do the data support the conclusions?

Reviewer #1: Partly

Reviewer #2: (No Response)

3. Has the statistical analysis been performed appropriately and rigorously? 

Reviewer #1: I Don't Know

Reviewer #2: (No Response)

4. Have the authors made all data underlying the findings in their manuscript fully available?

Reviewer #1: Yes

Reviewer #2: (No Response)

5. Is the manuscript presented in an intelligible fashion and written in standard English?

Reviewer #1: No

Reviewer #2: (No Response)

6. Review Comments to the Author

Reviewer #1: The authors have addressed my concerns and the manuscript has improved. However, reported baseline calorie intake is extremely surprising, being very hypocaloric in patients with obesity. I suggest to review records once again and elaborate on this aspect which seems extremely unlikely to be true. Moreover, English language needs extensive revision especially in the newly added parts.

Reviewer #2: (No Response)

7. PLOS authors have the option to publish the peer review history of their article (what does this mean?). If published, this will include your full peer review and any attached files.

Reviewer #1: No

Reviewer #2: No

---

## [Author Response · Author response to Decision Letter 1]

27 Aug 2021

Dear Sir, 

Thank you for your prompt response and further comments and suggestions. I hereby provide the details of the amendments as required.

Academic Editor:

The reviewers are mostly satisfied with the revised manuscript. Please comment on what seems to be common underreporting of energy intake in the discussion, as per Reviewer 1. If food was not provided to the study participants, please add as a limitation as well as the dietary reporting using 3-days diet diaries. 

Thank you so much for these suggestions. The points have been included as suggested.

Reviewer #1: The authors have addressed my concerns and the manuscript has improved. However, reported baseline calorie intake is extremely surprising, being very hypocaloric in patients with obesity. I suggest to review records once again and elaborate on this aspect which seems extremely unlikely to be true. 

Thank you for the comments. We have reviewed the records again and confirm the findings. However, we acknowledge the points raised and have included this in the discussion and limitation accordingly. 

This statement has been added. “With regards to the baseline macronutrients intakes, the notably low baseline calorie consumption could probably be due to considerable under-reporting and under-estimation, both intentionally and unintentionally, as similarly reported by a previous study [25].” 

References have been adjusted accordingly.

Moreover, English language needs extensive revision especially in the newly added parts.

We have sent the manuscript for proof reading. The text score was 82%. Corrections have been made accordingly.

Thank you again for your suggestions and comments. We look forward to your positive review.

---

## [Editor Report · Decision Letter 2]

29 Sep 2021

Safety and Efficacy of Very Low Carbohydrate Diet in Patients with Diabetic Kidney Disease- A Randomized Controlled Trial

PONE-D-21-17598R2

Dear Dr. Abdul Ghani,

We’re pleased to inform you that your manuscript has been judged scientifically suitable for publication and will be formally accepted for publication once it meets all outstanding technical requirements.

Kind regards,

Dorit Samocha-Bonet

Academic Editor

PLOS ONE
---

## [Editor Report · Acceptance letter]

4 Oct 2021

PONE-D-21-17598R2 

Safety and Efficacy of Very Low Carbohydrate Diet in Patients with Diabetic Kidney Disease- A Randomized Controlled Trial 

Dear Dr. Abdul Ghani:

I'm pleased to inform you that your manuscript has been deemed suitable for publication in PLOS ONE. Congratulations! Your manuscript is now with our production department. 

Kind regards, 

on behalf of

Dr. Dorit Samocha-Bonet 

Academic Editor

PLOS ONE